# Rivers as linear elements in landform evolution models

Stefan Hergarten[1]

[1]Institut für Geo- und Umweltnaturwissenschaften, Albertstr. 23B, 79104 Freiburg, Germany

**Correspondence:** Stefan Hergarten
(stefan.hergarten@geologie.uni-freiburg.de)

**Abstract.** Models of detachment-limited fluvial erosion have a long history in landform evolution modeling in mountain ranges. However, they suffer from a scaling problem when coupled to models of hillslope processes due to the flux of material from the hillslopes into the rivers. This scaling problem causes a strong dependence of the resulting topographies on the spatial resolution of the grid. A few attempts based on the river width have been made in order to avoid the scaling problem, but none of them appears to be completely satisfying. Here a new scaling approach is introduced that is based on the size of the hillslope areas in relation to the river network. An analysis of several simulated drainage networks yields a power-law scaling relation for the fluvial incision term involving the threshold catchment size where fluvial erosion starts and the mesh width. The obtained scaling relation is consistent with the concept of the steepness index and does not rely on any specific properties of the model for the hillslope processes.

## 1 Introduction

Fluvial incision is a major if not even dominant component of long-term landform evolution in orogens. When modeling fluvial erosion, restriction to the detachment-limited regime considerably simplifies the equations. Here it is assumed that the erosion rate at any point of a river can be predicted from local properties such as discharge and slope, while sediment transport is not considered. The generic differential equation for the topography $H(x_1, x_2, t)$ of a landform evolution model with detachment-limited fluvial erosion reads

$$\frac{\partial H}{\partial t} = U - E - \mathrm{div}\,\boldsymbol{q} \tag{1}$$

where $U$ is the uplift rate and $E$ the rate of fluvial incision. The third term describes a local transport process at the hillslopes where $\boldsymbol{q}$ is the flux density and div the 2D divergence operator. Linear diffusion is the simplest model here; it was considered in the context of landform evolution by Culling (1960) even before models of fluvial erosion came into play. However, there are also more sophisticated models for $\boldsymbol{q}$ taking into account the nonlinear dependencies of hillslope processes on topography (e.g., Andrews and Bucknam, 1987; Howard, 1994; Roering et al., 1999).

Concerning the fluvial incision term $E$, assuming a power-law function of the catchment size $A$ and the channel slope $S$,

$$E = KA^m S^n, \tag{2}$$

has become some kind of paradigm. The parameter $K$ is denoted erodibility. It is a lumped parameter subsuming all influences on erosion other than channel slope and catchment size, so that it is not a property of the rock alone, but also depends on climate in a nontrivial way (e.g., Ferrier et al., 2013; Harel et al., 2016).

Equation (2) is often called stream power approach since it can be interpreted in terms of energy dissipation of the water per channel bed area if an empirical relationship between channel width and catchment size is used (e.g., Whipple and Tucker, 1999). However, the idea behind this approach even dates back to the empirical study of longitudinal channel profiles by Hack (1957). In this study, a power-law relationship between channel slope and drainage area was found, often called Flint's law (Flint, 1974). This relationship is nowadays usually written in the form

$$S = k_{\mathrm{s}} A^{-\theta} \tag{3}$$

where $\theta$ is the concavity index and $k_{\mathrm{s}}$ the steepness index. Assuming that Eq. (3) is the fingerprint of spatially uniform steady-state conditions, it predicts $\frac{m}{n} = \theta$ and allows for a convenient interpretation of the erodibility. If local transport (last term in Eq. 1) is neglected, the steepness index follows the relation

$$k_{\mathrm{s}}^n = \frac{E}{K}. \tag{4}$$

This relation allows for a simple adjustment of the lumped parameter $K$ in such a way that a given channel steepness is achieved at a given erosion rate.

## 2  The scaling problem

While widely used and in principle simple, all models of the type described by Eqs. (1) and (2) suffer from a scaling problem. Mathematically, the problem is that catchment sizes are not well-defined in the continuum limit as the catchment of each point degenerates to a line. When considered on a discrete grid, rivers are represented as linear objects with a width of one pixel. Thus, the total surface area of the pixels covering the network of the large rivers decreases with decreasing mesh width.

If local transport is not considered, the scaling problem leads to a canyon-like topography where the width of the valleys decreases with mesh width. This behavior is illustrated in Figs. 1 and 2 where two steady-state topographies with mesh widths of $\delta = 0.01$ ($100 \times 100$ nodes) and $\delta = 0.002$ ($500 \times 500$ nodes) are considered. All parameter values are set to unity except for $m = 0.5$, so that $\theta = 0.5$. The northern and southern boundaries are held at zero elevation, while the western and eastern boundaries are periodic. The topographies were obtained from the landform evolution model OpenLEM that was used in some previous studies (e.g., Robl et al., 2017; Wulf et al., 2019), but has not been published explicitly. It uses the D8 flow routing scheme (O'Callaghan and Mark, 1984) and a fully implicit scheme (Hergarten and Neugebauer, 2001; Hergarten, 2002), so that large time steps can be performed in order to ensure that a steady state is achieved. The simulation on the fine grid was started from a flat topography with a small random disturbance, while the simulation on the coarse grid was started from a downsampled version of the finer topography.

Relief increases with decreasing grid spacing because the smallest catchment size that can be resolved is $A_{\mathrm{min}} = \delta^2$, and maximum equilibrium slope is proportional to $A_{\mathrm{min}}^{-\theta} = \delta^{-2\theta}$ according to Eq. (3). As nodes with small catchment sizes can

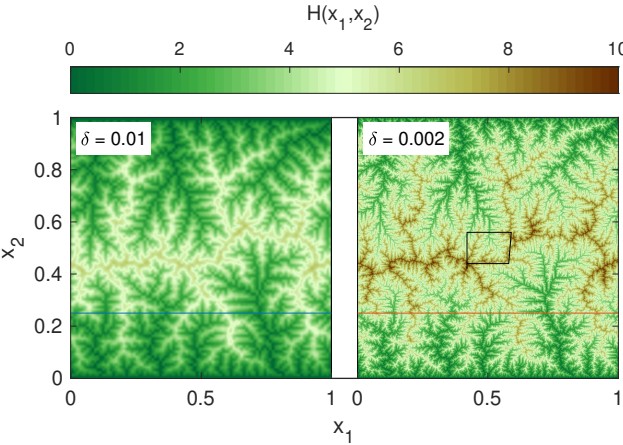

**Figure 1.** Fluvial equilibrium topographies computed for identical parameter values on grids with different spacing ($\delta = 0.01$, $100 \times 100$ nodes and $\delta = 0.002$, $500 \times 500$ nodes). The horizontal lines refer to the profiles analyzed in Fig. 2, and the rectangle marks the region shown in Fig. 4.

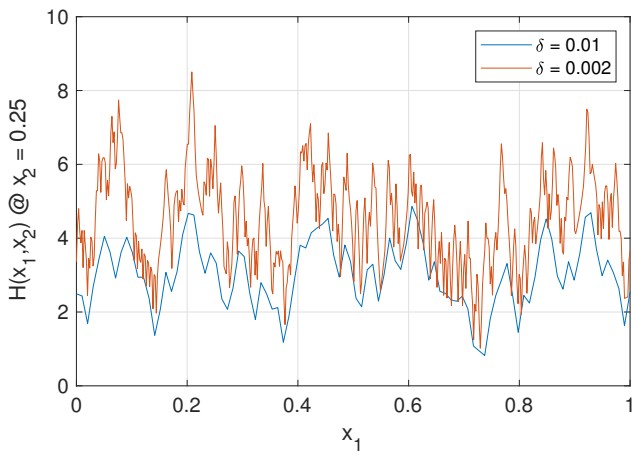

**Figure 2.** Profiles through the topographies shown in Fig. 1.

drain directly into large rivers, this increase is not restricted to major drainage divides, but also result in steep valley flanks. The heights of the valley floors are, however, hardly affected by the spatial resolution. Catchment sizes of large rivers even converge in the limit $\delta \to 0$, so that longitudinal profiles of large rivers become stable for $\delta \to 0$ according to Eq. (3). Thus, relief and also mean elevation depend on the spatial resolution for the simplest model without local transport, while large rivers

are hardly affected.

The independence of river steepness of resolution is, however, lost as soon as local transport comes into play. Figure 3 shows the example of short, parallel river segments with unit spacing (periodic in $x_2$ direction) in equilibrium with constant uplift.

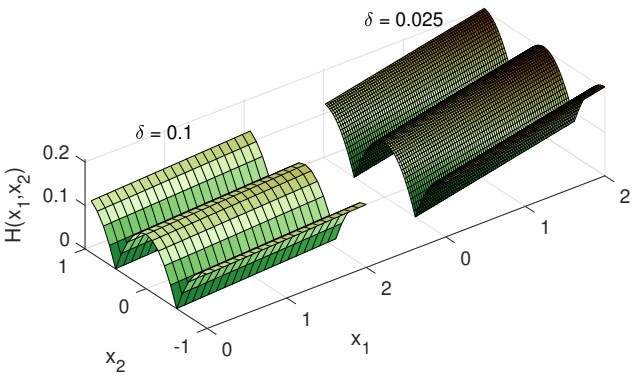

**Figure 3.** River segments in equilibrium with uplift for different mesh widths $\delta$.

Linear diffusion

$$\boldsymbol{q} = -D\nabla H \tag{5}$$

was assumed as the simplest model for local transport. As in the previous example, all parameters except for $m = 0.5$ were set to unity. A catchment size of $A = 10^6$ was assumed for each river segment, so that the channel slope should theoretically be $S = 10^{-3}$ in equilibrium with $U = 1$. While the topography of the hillslopes is in principle independent of the grid spacing $\delta$, the river segment becomes steeper if $\delta$ decreases.

The reason for the increasing channel steepness is that the local transport is conservative, so that the river does not only have to incise into the rock at its bed, but also has to remove the material coming from the hillslopes. Regardless of the model used for local transport, a flux of $(d - \delta)U$ per river length enters the site that contains the river in equilibrium where $d$ is the valley spacing. Then the discretized divergence of the flux density is

$$\mathrm{div}\boldsymbol{q} = -\frac{(d - \delta)U}{\delta}. \tag{6}$$

Inserting this result into the steady-state version of Eq. (1) yields

$$E = U - \mathrm{div}\boldsymbol{q} = \frac{d}{\delta}U, \tag{7}$$

so that the fluvial erosion rate required for compensating uplift is by a factor $\frac{d}{\delta}$ higher than it would be without local transport. This requires an increase in channel slope by a factor of $\left(\frac{d}{\delta}\right)^{\frac{1}{n}}$ according to Eq. (2).

This scaling issue has been known for more than 25 years, and two approaches have been suggested to overcome the problem were proposed. Howard (1994) suggested a subpixel representation of the rivers where a river segment only covers a fraction of a grid cell. It was assumed that this fraction is $\frac{w}{\delta}$ where $w$ is the river width, and then the fluvial incision term $E$ was multiplied with this factor. Perron et al. (2008) transferred this concept to the detachment-limited case. According to Eq. (7), rescaling $E$ by the factor $\frac{w}{\delta}$ yields

$$E = \frac{d}{w}U, \tag{8}$$

so that the dependency on $\delta$ indeed vanishes.

While straightforward at first sight, this scaling approach is not free of problems. The channel width in general increases in downstream direction, so that equilibrium river profiles are no longer consistent with Eq. (3). Perron et al. (2008) avoided this problem by assuming a constant channel width and postponing it to subsequent studies. As discussed by Pelletier (2010), taking into account an increase of channel width in downstream direction would require a reduction of the exponent $m$ in Eq. (2) in order to keep it consistent with Eq. (3). However, unit and meaning of the erodibility $K$ would change then.

In order to overcome this problem, Pelletier (2010) suggested to leave the fluvial incision term as is and rescale the local transport term div$\boldsymbol{q}$ by the inverse factor $\frac{\delta}{w}$ at sites containing rivers. Practically, this rescaling means that the flux of material coming from the hillslopes is not distributed over the entire grid cell, but only over the part of the area covered by the river. So it can be seen as the inverse of the subpixel approach of Howard (1994) and Perron et al. (2008) applied to the local transport instead of the fluvial erosion. For the steady-state example considered above, this rescaling leads to

$$\mathrm{div}\boldsymbol{q} = -\frac{(d-\delta)U}{w} \tag{9}$$

instead of Eq. (6), so that

$$E = U - \mathrm{div}\boldsymbol{q} = \frac{(d+w-\delta)}{w}U. \tag{10}$$

For $w \ll d$ and $\delta \ll d$, however, this relation approaches Eq. (8), so that this concept suffers from the same problem as the approach of Howard (1994) and Perron et al. (2008).

So there seems to be no completely satisfying solution of the scaling problem so far. Several contemporary modeling studies (e.g., Duvall and Tucker, 2015; Gray et al., 2018; Wulf et al., 2019; Reitman et al., 2019) use neither of the two approaches, but implement Eq. (1) as is without taking its dependence on the grid scale into account. This is not a crucial problem as long as simulations with different spatial resolutions are not compared and as long as we are aware that the erodibility $K$ has a limited meaning. As soon as the relevance of fluvial erosion and hillslope processes is assessed quantitatively or scaling relations are

developed (e.g., Theodoratos et al., 2018), the problem may become crucial. A further discussion is given in Sect. 5.

  Other recent approaches navigate around the scaling problem by neglecting the flux of material from the hillslopes into the rivers. The recently presented landform evolution model TTLEM (Campforts et al., 2017) makes a distinction by catchment size in such a way that fluvial erosion only acts on sites with a catchment size above a given threshold $A_{\mathrm{c}}$, while hillslope processes only act at smaller catchment sizes. It is assumed that all hillslope material entering the rivers is immediately excavated without

any further effect, so that fluxes from hillslopes into rivers can be disregarded, and the scaling problem does not occur. This approach reduces the interaction between rivers and hillslopes to a one-way coupling where only the rivers have an influence on the evolution of the hillslopes and can be seen as an implementation of bedrock incision in the strict sense. While it seems that the terms detachment-limited erosion and bedrock incision are sometimes used synonymously, it should be clarified that the applicability of the concept of pure bedrock incision is probably much narrower than that of detachment-limited erosion,

in particular if highly resistant material is brought into the channels (Shobe et al., 2016). The same in principle holds for the model most widely used in the context of drainage divide migration (Goren et al., 2014) where analytical solutions for hillslope processes are used on the sub-pixel scale.

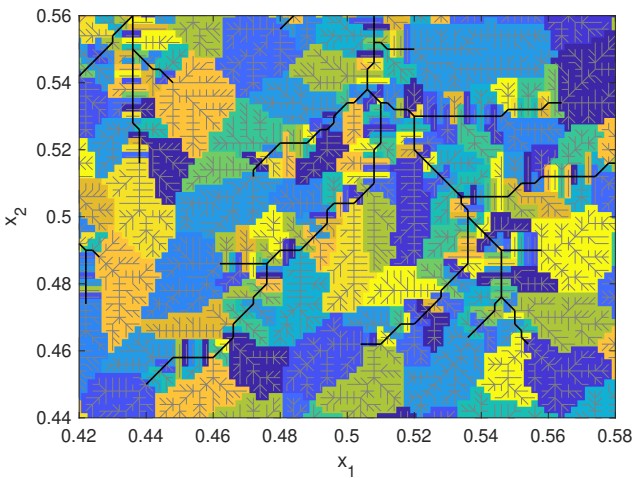

**Figure 4.** Flow pattern of the central region of Fig. 1. Black lines show rivers with $A \geq A_\mathrm{c}$ for $A_\mathrm{c} = 100$ pixels. Gray lines are channels with $A < A_\mathrm{c}$ considered as hillslope sites. Each colored area consists of one channel site plus the hillslope area that drains to this site without passing another upstream chnanel site.

## 3   A new scaling approach

The simple example considered in the previous section involves a dependence on grid spacing $\delta$ according to the factor $\frac{d}{\delta}$ without rescaling (Eq. 7). Both approaches for rescaling replace the dependence on $\delta$ by a dependence on the channel width $w$, so that a factor $\frac{d}{w}$ remains (Eq. 8). This is, however, still a problem if $w$ is not constant. The occurrence of the factor $\frac{d}{w}$ suggests that the valley spacing $d$ would be a more suitable characteristic length scale for rescaling than $w$ if we want to preserve the form of the erosion law (Eq. 2) without changing the exponents $m$ and $n$. In the following, a concept that generalizes the simple example of parallel rivers to dendritic networks is developed.

Let us start from the simplest approach to distinguish channel sites from hillslopes by defining a threshold catchment size $A_\mathrm{c}$ in such a way that all sites with $A \geq A_\mathrm{c}$ are river segments, while all sites with $A < A_\mathrm{c}$ belong to hillslopes. As local transport is conservative, all material eroded anywhere has to be removed by the river sites, so that we need to know how much material each river sites receives from the hillslopes. The area of the respective hillslopes can be determined for a given topography without any specific assumptions on the transport process except for the direction of transport. The simplest model is to assume that local transport follows the hypothetic channel network at the hillslopes, i.e., the direction of steepest descent on a purely fluvial topography. Figure 4 illustrates this concept. Each colored area consists of one channel site and the hillslope area that delivers its eroded material to this site, i.e, of those sites that drain to the considered site without passing any other upstream channel site.

If the size of this area was the same for each river site, rescaling the fluvial erosion rate (Eq. 2) according to

$$E = A_\mathrm{e} K A^m S^n \tag{11}$$

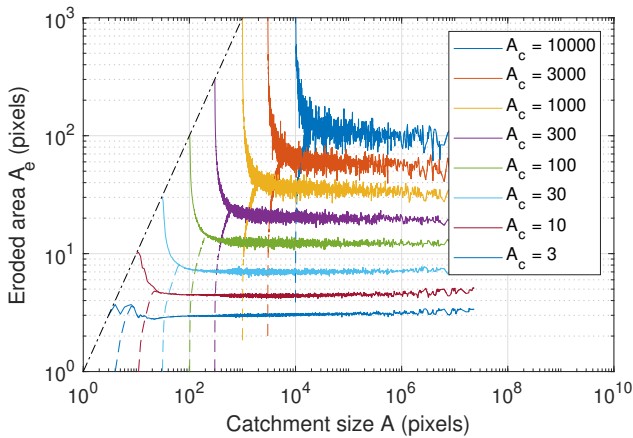

**Figure 5.** Eroded area $A_e$ as a function of the catchment size $A$ for different fluvial thresholds $A_c$. Raw data were used for those catchment sizes that occurred at least 1000 times on the grid. Otherwise, data were binned dynamically so that there are at least 1000 points in each bin.

where $A_e$ is the size of this area measured in DEM pixels (i.e., the number of sites) would already solve the scaling problem. However, it is immediately recognized in Fig. 4 that the sizes of these areas are highly variable. A random variation in these sizes is not a problem. If $A_e$ in Eq. (11) is the mean size, channel steepness will just vary randomly, which is also found in nature. A systematic dependence of $A_e$ on catchment size would, however, be a problem. In this case, equilibrium river profiles
would be no longer consistent with Eq. (3), so that the problem would be basically the same as in the previous approach for a non-constant channel width.

In the following, numerically obtained equilibrium drainage networks are analyzed in order to find out how $A_e$ depends on $A$ and on $A_c$. More precisely, $A_e$ is the mean size of all hillslopes areas draining to channel sites with a given catchment size $A$ at a given fluvial threshold $A_c$ (plus the respective channel site). For simplicity, all areas are measured in DEM pixels in the
following considerations, i.e., as a number of sites. Starting point of the analysis is the drainage network of a fluvial equilibrium topography on a square $L \times L$ grid with $L = 10000$. Boundary conditions and parameter values except for the grid size are the same as in the smaller examples shown in Fig. 1.

Figure 5 reveals that the eroded area $A_e$ increases with the fluvial threshold $A_c$, but becomes independent of $A$ if the catchment size $A$ is sufficiently large. This means that the hillslopes draining to large rivers are not systematically larger than
those draining to small rivers. It is the reason why we will arrive at a scaling relation that preserves the form of Eq. (2) and avoids the problem occurring if the river width is used for scaling.

The increase of $A_e$ if $A$ approaches $A_c$ can be explained by distinguishing between river segments and channel heads. Let us define channel heads as those sites without any tributary with $A \geq A_c$, i.e., as those sites that are only supplied by hillslopes. All other sites with $A \geq A_c$ are considered as river segments. All sites with $A = A_c$ are channel heads and thus follow the
relation $A_e = A$, so that all curves start at the dotted line in Fig. 5. The resulting values $A_e$ of the river segments (without the channel heads) are shown by the dashed lines in Fig. 5. The increase of $A_e$ if $A$ approaches $A_c$ even turns into a decrease then.

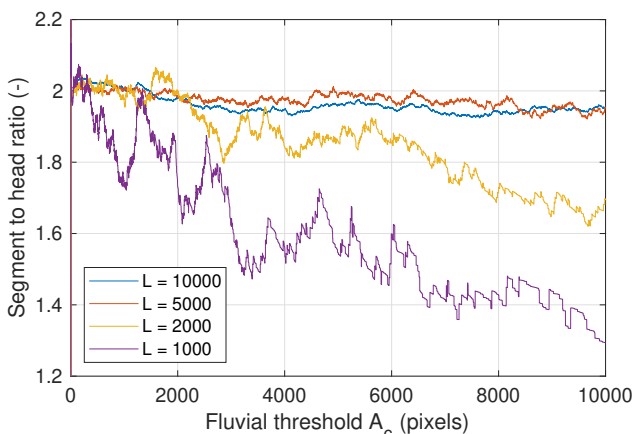

**Figure 6.** Ratio of total area eroded by all river segments and total area eroded by all channel head sites as a function of the fluvial threshold $A_\mathrm{c}$.

This decrease arises from the limitation $A_\mathrm{e} \leq A - A_\mathrm{c}$ that holds for all river segments as those have at least one tributary cell contributing at least $A_\mathrm{c}$. So the contribution of the hillslopes must be small if $A$ is only slightly larger than $A_\mathrm{c}$. However, the decrease is exaggerated by the logarithmic scale and concerns only a small number of sites. So it makes sense to assume that $A_\mathrm{e}$ is independent of $A$ for river segments.

Both the number of river segment sites and the number of channel head sites decrease with increasing threshold $A_\mathrm{c}$. The decrease of the latter is faster, so that the ratio of the numbers of head sites vs. river sites converges to zero for large $A_\mathrm{c}$. This is, however, not true for the total contributions. Figure 6 shows the ratio of the sum of the $A_\mathrm{e}$ values of all river segments and the sum of the $A_\mathrm{e}$ values of the channel heads. It can also be interpreted as the ratio of the total area that must be eroded by the river segments over the total area that must be eroded by the channel heads. The results shown for different grid sizes shown in Fig. 6 suggests that this ratio becomes constant in the limit of large grid sizes. It apparently approaches a value of about 2 here, which means that the river segments contribute about two thirds to total fluvial erosion and the channel heads one third.

This result suggests that the dependency of $A_\mathrm{e}$ on the threshold $A_\mathrm{c}$ is determined by the cumulative distribution $P(A)$ of the catchment sizes in the drainage network. This distribution describes the probability that a randomly selected site has a catchment size $\geq A$. The probability $P(A_c)$ evaluated at the fluvial threshold is the ratio of the area covered by all channel pixels and the total area. It can be interpreted as a drainage density (river length per total area) on a discrete grid. Then a fraction $P(A_\mathrm{c})$ of the considered domain must erode a given fraction (here about two thirds) of the domain, leading to the relation

$$A_\mathrm{e} = \frac{\gamma}{P(A_\mathrm{c})} \tag{12}$$

with $\gamma \approx \frac{2}{3}$ for this network. While $A_\mathrm{e}$ can be measured directly for the considered drainage network, its relation to $P(A)$ (Eq. 12) is useful as this distribution has already been investigated in several studies on natural and modeled drainage networks (Rodriguez-Iturbe et al., 1992a; Maritan et al., 1996b; Rodriguez-Iturbe and Rinaldo, 1997; Rinaldo et al., 1998; Hergarten and

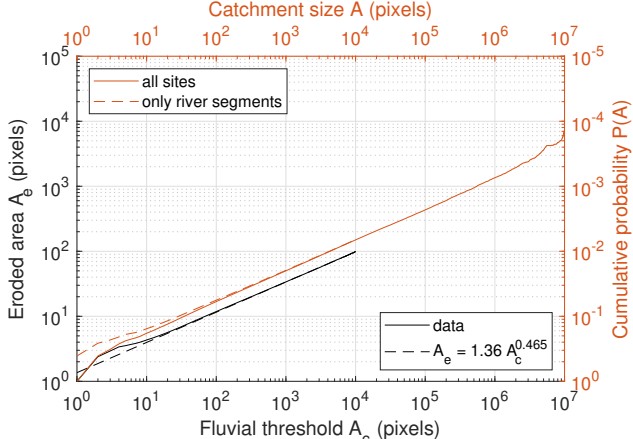

**Figure 7.** Black axes: eroded area as a function of the fluvial threshold. Red axes: cumulative distribution of the catchment sizes.

Neugebauer, 2001; Hergarten, 2002; Hergarten et al., 2014, 2016). It was found that $P(A)$ follows a power-law distribution

$$P(A) \sim A^{-\beta} \tag{13}$$

over a reasonable range where a range $\beta \in [0.41, 0.46]$ was found except for the two latest studies. In these studies, larger
networks were considered making use of increasing data availability and computing capacities. An exponent very close to
0.5 was found for both optimal channel networks (OCNs, see below) (Hergarten et al., 2014) and a real river pattern at the
continental scale (Hergarten et al., 2016).

Equations (12) and (13) suggest a power-law relation

$$A_{\mathrm{e}} = \alpha A_{\mathrm{c}}^{\beta} \tag{14}$$

between the eroded area and the fluvial threshold. The validity of Eqs. (12), (13), and (14) is investigated in Fig. 7. Comparing
the two solid curves reveals that Eq. (12) does not hold exactly since the curves come closer to each other for decreasing
catchment sizes. The reason is that $A_{\mathrm{e}}$ only refers to the river segments without the channel heads, so that $P(A_{\mathrm{c}})$ in Eq. (12)
should also exclude the channel head sites. The dashed red line in Fig. 7 showing the accordingly reduced distribution $P(A)$
illustrates that Eq. (12) indeed holds then, and that the effect vanishes for large $A_{\mathrm{c}}$.

The black dashed line in Fig. 7 refers to the best-fit power-law relation according to Eq. (14). It is based on all integer
values of $A_{\mathrm{c}}$ from 1 to 10,000 assuming equal errors, so that the large values of $A_{\mathrm{c}}$ practically have a high weight in the fit.
The power law with the obtained values $\alpha = 1.360$ and $\beta = 0.465$ fits the data well with a relative error of less than 5 % for
$A_{\mathrm{c}} \in [15, 10000]$ and less than 1 % for $A_{\mathrm{c}} \in [400, 10000]$. The deviations are larger for smaller fluvial thresholds due to the fact
that dendritic networks cannot be represented well on a regular lattice at small scales.

The relation to the catchment-size distribution (Eqs. 12 and 13) suggests that the power-law dependency of $A_{\mathrm{e}}$ on $A_{\mathrm{c}}$ (Eq. 14)
should be universal. For testing this hypothesis, a set of equilibrium topographies with $\theta \in \{0.25, 0.45, 0.5, 0.75\}$ was analyzed.

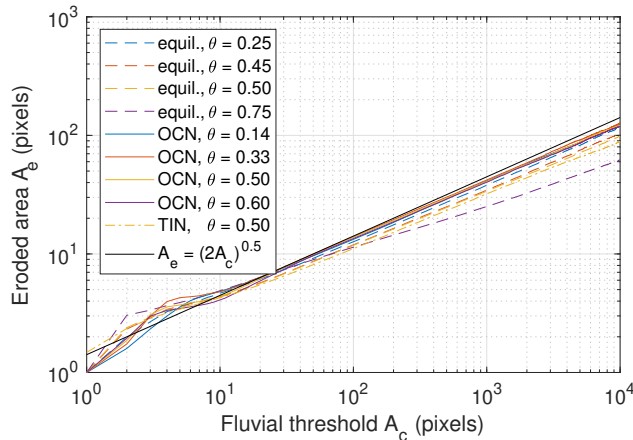

**Figure 8.** Eroded area $A_e$ as a function of the fluvial threshold $A_c$ for the considered drainage networks. For clarity, only the results obtained from the largest domains are plotted.

These values cover the range that has been found so far under relatively homogeneous conditions (e.g., Robl et al., 2017). The value $\theta = 0.45$ was added as it is often used as a reference value instead of $\theta = 0.5$ (e.g., Whipple et al., 2013; Lague, 2014). Parameter values and boundary conditions are the same as in the previous example. Since the exponent $n$ has no immediate effect on equilibrium topographies, values $n \neq 1$ were not considered.

The power-law parameters $\alpha$ and $\beta$ obtained from equilibrium topographies on different lattice sizes $L$ are given in Table 1. In addition, the original data for the largest grids are shown in Fig. 8. The results are overall similar with a tendency to lower exponents $\beta$ for increasing $\theta$. A notable deviation is only found for the very high concavity index $\theta = 0.75$. Here the slopes become very steep at small catchment sizes, resulting in a slower migration of drainage divides during the simulation (Robl et al., 2017). As a result, the topography reaches a steady state quite soon, so that there is finally less reorganization in the drainage network with regard to the initial random pattern. In this sense, the lower exponents found for $\theta = 0.75$ can be seen as some fingerprint of poorly organized river patterns, but are probably not relevant for the rivers that were the empirical basis of the stream power law. These findings confirm that the concavity index $\theta$ has a minor effect on the topology of the drainage networks, although it strongly affects the shape of longitudinal river profiles and thus the topography.

In addition, Table 1 and Fig. 8 also contain results obtained from optimal channel networks (OCNs) on a grid with $L = 4096$. Optimal channel networks are derived from the principle of minimum energy dissipation and have been widely used in the context of river networks (e.g., Howard, 1990; Rodriguez-Iturbe et al., 1992c, b; Rinaldo et al., 1992; Maritan et al., 1996a, b; Rinaldo et al., 1998). The networks considered here are those shown in Fig. 1 of Hergarten et al. (2014) where $\theta$ is related to the parameter $n$ used there by $\theta = \frac{n-1}{n+1}$. The values of $A_e$ of OCNs are overall slightly higher than those of the equilibrium topographies, and the variation with $\theta$ is lower. As OCNs are organized more strongly than drainage networks of arbitrary equilibrium topographies, the lower variability among OCNs is not surprising.

**Table 1.** Parameter values of the power-law relation between eroded area and fluvial threshold (Eq. 14) obtained from different simulated drainage networks on regular lattices with $L \times L$ nodes.

| | $\theta$ | $L$ | $\alpha$ | $\beta$ |
|---|---|---|---|---|
| | | 5000 | 1.264 | 0.492 |
| | 0.25 | 2000 | 1.072 | 0.511 |
| | | 1000 | 1.587 | 0.470 |
| steady-state topographies | | 5000 | 1.273 | 0.478 |
| | 0.45 | 2000 | 1.586 | 0.451 |
| | | 1000 | 1.047 | 0.499 |
| | | 10,000 | 1.360 | 0.465 |
| | | 5000 | 1.434 | 0.459 |
| | 0.50 | 2000 | 1.807 | 0.423 |
| | | 1000 | 1.579 | 0.440 |
| | | 10,000 | 1.653 | 0.393 |
| | | 5000 | 1.715 | 0.388 |
| | 0.75 | 2000 | 1.433 | 0.412 |
| | | 1000 | 2.179 | 0.359 |
| OCNs | 0.14 | | 1.487 | 0.480 |
| | 0.33 | 4096 | 1.626 | 0.473 |
| | 0.50 | | 1.508 | 0.478 |
| | 0.60 | | 1.521 | 0.475 |

Table 2 provides additional results obtained from steady-state topographies on triangulated irregular networks (TINs). Numbers of neighbors, distances to neighbors, and areas of pixels are variable here. The latter are defined by the Voronoi diagram. Nondimensional areas (in DEM pixels) are normalized to the mean pixel size given by $\delta^2 = \frac{A_{tot}}{N}$ where $A_{tot}$ is total area and $N$ the number of nodes. The values listed in the Table 2 and the respective curve in Fig. 8 show that the results obtained from TINs are close to those obtained from regular meshes.

These results suggest to define the values $\alpha = 1.508$ and $\beta = 0.478$ obtained from the OCN with $\theta = 0.5$ as reference values. The question is, however, whether such a precision is useful for applications. In particular, $\beta = 0.5$ would be more convenient than lower values. In the considerations made above, all areas are measured in DEM pixels and are thus nondimensional properties. Considering $A_c$ as a physical (dimensional) area, $A_c$ has to be replaced by $\frac{A_c}{\delta^2}$ in Eq. (14). Then the fluvial erosion rate (Eq. 11) turns into

$$E = \alpha \left( \frac{A_c}{\delta^2} \right)^{\beta} K A^m S^n, \tag{15}$$

**Table 2.** Parameter values of the power-law relation between eroded area and fluvial threshold (Eq. 14) obtained from different simulated drainage networks on triangular lattices with $N$ nodes for $\theta = 0.5$.

| $N$ | $\alpha$ | $\beta$ |
|---|---|---|
| $2 \times 10^7$ | 1.630 | 0.433 |
| $1 \times 10^7$ | 1.611 | 0.435 |
| $5 \times 10^6$ | 1.264 | 0.466 |
| $2 \times 10^6$ | 1.332 | 0.454 |
| $1 \times 10^6$ | 1.400 | 0.445 |
| $5 \times 10^5$ | 1.432 | 0.450 |

so that the fluvial incision term scales like $\delta^{-2\beta}$. For $\beta = 0.5$, the fluvial term scales like $\frac{1}{\delta}$. This is not only convenient, but also leads to basically the same scaling relation assumed by Perron et al. (2008). The only difference is that the term $\alpha\sqrt{A_c}$ occurring here was interpreted as a channel width $w$ and then assumed to be constant for all rivers, so that it lost its physical meaning. So the new formulation of the fluvial incision term also fixes the concern raised by Pelletier (2010) that led to the alternative formulation where the hillslope transport term was rescaled.

In order to estimate $\alpha$ for $\beta = 0.5$, it is helpful to know in which region of Fig. 8 we are in typical model applications. A breakdown of Flint's law (Eq. 3) was reported at catchment sizes between between about 0.1 km$^2$ and 5 km$^2$ (Montgomery and Foufoula-Georgiou, 1993; Stock and Dietrich, 2003; Wobus et al., 2006). However, channel steepness declines at small catchment sizes, so that this breakdown rather implies that other erosion processes come into play than that fluvial erosion is no longer active. In turn, many small springs in mountain regions have discharges in the order of magnitude of 0.1 liters per second (e.g., Hergarten et al., 2016), corresponding to catchment sizes $A < 0.01$ km$^2$, but it is not clear whether the erosive action of the resulting small streams follows Flint's law. Reasonable estimates of $A_c$ are probably between these two ranges. Assuming a spatial resolution of about 100 m or a bit less, $A_c$ will be in the order of magnitude of a few to 100 DEM pixels. As illustrated by the black line in Fig. 8, $\alpha = \sqrt{2}$ provides a reasonable estimate for this range with simple numbers as $\alpha A_c^\beta = \sqrt{2A_c}$ then. With this estimate, the scaling factor for the fluvial erosion rate is $\frac{\sqrt{2A_c}}{\delta}$, and the modified stream-power law for fluvial erosion turns into

$$E = \frac{\sqrt{2A_c}}{\delta} K A^m S^n. \tag{16}$$

## 4 Numerical examples

Let us first return to the example of parallel rivers considered in Fig. 3. It was found in Sect. 2 that the topography of the hillslopes was robust against the spatial resolution, while the channel slope increases with decreasing grid spacing $\delta$. Both approaches previously published fix this problem, but the channel slopes are by a factor of $\frac{d}{w}$ too steep compared to what is expected from the erodibility.

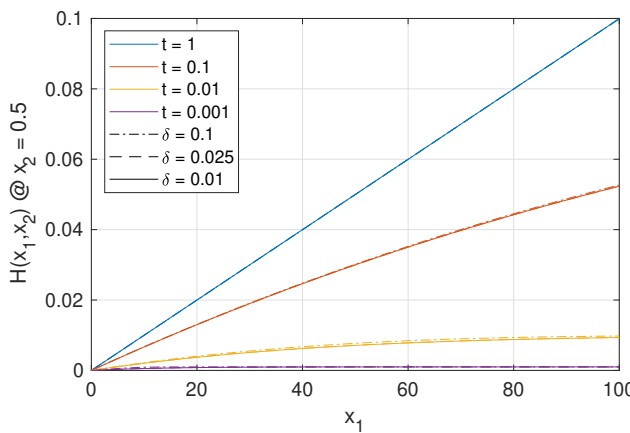

**Figure 9.** Numerical results for the scenario considered in Fig. 3. The river profiles obtained for $\delta = 0.025$ and $\delta = 0.01$ cannot be distinguished visually.

It should be noted that this example is not related to the approach to estimate $A_\mathrm{e}$ from $A_\mathrm{c}$ for dendritic networks (Eqs. 15 and 16), but can only test the validity of the principal scaling approach (Eq. 11). The size of the area $A_\mathrm{e}$ does not follow Eq. (14), but is defined by the geometry as $A_\mathrm{e} = \frac{d}{\delta}$ (measured in DEM pixels). Figure 9 shows the numerical results for the parameter values used in Fig. 3 for different values of $\delta$. The simulation was started from a flat topography where the flow paths of the parallel rivers are predefined. As the problem is linear for $n = 1$, this example can also be seen as the change in the river profile through time if uplift suddenly increases at $t = 0$, while the base level remains constant. The results show that the equilibrium profile achieved for large times is reproduced correctly, and that the time-dependent behavior is also robust against the resolution. This means that the scaling approach itself (Eq. 11) yields both the correct equilibrium behavior and the correct time scale.

The second example refers to the scenario considered in Fig. 1, but extended by a fluvial threshold $A_\mathrm{c} = 10^{-5}$ and by linear diffusion with a diffusivity $D = 10^{-5}$. The threshold $A_\mathrm{c}$ is a property of the fluvial erosion process, while the diffusive hillslope process is not related to it. It is thus assumed that fluvial erosion acts only at sites where $A \geq A_\mathrm{c}$, while diffusion is active everywhere. A TIN representation is used in order to avoid artefacts from the combination of the eight-neighbor (D8) flow routing scheme with the standard four-neighbor diffusion scheme on a regular mesh. The simulations are started from an almost flat topography with unit uplift. Uplift is switched off at $t = 50$ in order to observe the decay of the topography.

The mean steepness index $k_\mathrm{s}$ of the large rivers is plotted as a function of time in Fig. 10. Large rivers are defined by $A \geq 10^{-3}$ here, which is considerably larger than $A_\mathrm{c}$, but much smaller than the domain. As expected, the simulations performed without any rescaling of the erodibility (dashed lines) are strongly affected by the spatial resolution. The steepness index increases with increasing number of nodes $N$, i.e., with decreasing pixel size. In turn, the results obtained using the simple scaling relation (Eq. 16, solid lines) have a much weaker dependence on resolution. There is, however, a residual variation in

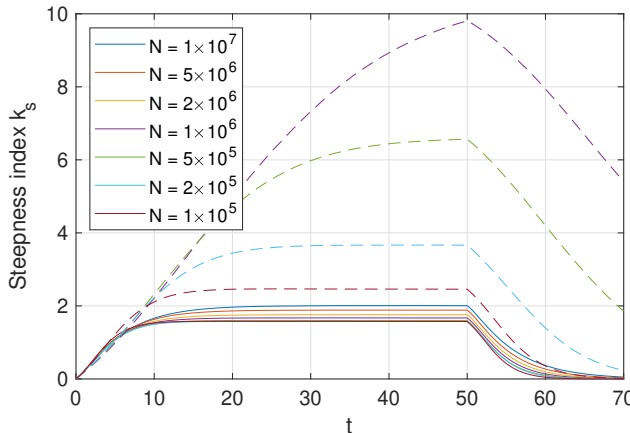

**Figure 10.** Mean steepness index $k_s$ of the large rivers obtained from simulations on TINs with different resolutions, defined by the total number of nodes $N$. Solid lines refer to the simplified scaling approach suggested in this paper (Eq. 16), while dashed lines refer to simulations performed without any rescaling. The latter are plotted only for $N \leq 10^6$.

channel steepness. The mean value of $k_s$ varies between about 1.6 and 2.0 over the considered range from $N = 10^5$ to $N = 10^7$. This result does not change fundamentally if a higher or lower threshold than $A \geq 10^{-3}$ is used for defining large rivers.

## 5 Discussion

It may be surprising that the example of fluvial incision and hillslope diffusion considered in the previous section yields a mean steepness index greater than one, although the scaling concept was developed in order to preserve channel steepness. The concept is, however, based on a generic hillslope process where the direction of transport follows a hypothetic fluvial equilibrium pattern and turns into fluvial erosion at a given threshold catchment size $A_c$. It is questionable whether any hillslope process occurring in nature comes close to this simple model. In the example considered here, the diffusion process is characterized by a diffusivity $D$ and is not related to $A_c$. The fluvial domain is affected by diffusion more and more with increasing diffusivity. As a consequence, slopes of small channels decrease, so that they erode less efficiently. This has to be compensated by the larger rivers, so that they must become steeper.

This is, however, a real property of the hillslope process here, and it is not the goal of the scaling approach to remove it. The concept presented here aims at removing the dependence on the resolution and to provide a way how values of the erodibility have to be interpreted. Here it is suggested that they should be considered in combination with a fluvial threshold $A_c$ in such a way that they would yield the expected channel steepness if the generic hillslope model was valid.

In turn, the residual dependence of channel steepness on resolution is a problem, in particular because it is not clear whether it converges in the limit $\delta \to 0 \, (N \to \infty)$. The problem arises from network reorganization which also affects the fluvial region. Diffusion disturbs the dendritic topology towards parallel flow where the model based on Hack's findings (Eq. 2) is not valid. Using an improved flow routing scheme that is able to distinguish channelized flow from parallel flow as suggested by Pelletier

(2010) and letting $A_\mathrm{c}$ self-adjust might reduce the problem. However, the aim of this study is to develop a simple, quite universal rescaling approach that avoids or at least reduces the dependence on resolution without modifying the applied model seriously. In this sense, Eq. (16) should be a good tradeoff.

Nevertheless it is important to keep the difference between detachment-limited erosion and pure bedrock incision in mind. Here it is assumed that the ability of the river to take up particles and carry them away concerns both the river bed and material coming from adjacent hillslopes. If we, conversely, assume that all material coming from the hillslopes is instantaneously removed by the river without any consequences, there is no feedback of the hillslopes to the rivers, and Eq. (1) does not require any rescaling.

The results of this study have consequences for scaling relations in coupled models of rivers and hillslopes. Theodoratos et al. (2018) conducted a comprehensive analysis of the problem with linear diffusion without rescaling. The parameters they used were the same as in the previous example (Fig. 10), so that it is immediately clear that their numerical results strongly depend on resolution. The authors argued that, following the approach of Pelletier (2010), both grid spacing and channel width are rescaled, so that the ratio $\frac{\delta}{w}$ remains constant, and the scaling issue is consistent throughout all scales. However, the results presented here show that the property relevant for compensating $\delta$ is not channel width, but $A_\mathrm{e}$ and thus $A_\mathrm{c}$. These parameters are, however, physical properties of the erosion process, so that they do not scale with the size of the domain. As a consequence, the characteristic horizontal length scale of the coupled system should rather be

$$l_\mathrm{c} = \frac{D}{\sqrt{A_\mathrm{c}} K} \qquad (17)$$

for $m = 0.5$ and $n = 1$ instead of $l_\mathrm{c} = \sqrt{\frac{D}{K}}$ used by Theodoratos et al. (2018). This problem also affects the recent extension by an erosion threshold (Theodoratos and Kirchner, 2020).

## 6 Conclusions

This study presents a simple scaling relation for the fluvial incision term in landform evolution models involving detachment-limited fluvial erosion and hillslope processes. In order to avoid a dependence of the simulated topographies on the spatial resolution of the grid, the fluvial incision term must be multiplied by a scaling factor depending on the ratio of the threshold catchment size $A_\mathrm{c}$ where fluvial erosion starts and the pixel size $\delta^2$ of the grid. The analysis of several simulated drainage networks yields a power-law dependence of the scaling factor in Eq. (15) with an exponent slightly lower than 0.5. However, for application in numerical models, a simpler approximation where the fluvial erosion rate is rescaled by a factor $\frac{\sqrt{2A_\mathrm{c}}}{\delta}$ is suggested. As this relation assumes a simple, generic hillslope process, it cannot provide an exact solution for all types of hillslope processes. In combination with such processes, e.g., diffusion, the dependence on the spatial resolution is not completely removed. Nevertheless, the simple scaling relation appears to be a reasonable tradeoff between accuracy and simplicity.

*Code and data availability.* All codes and computed data can be downloaded from the FreiDok data repository at **STILL WAITING FOR THE REPOSITORY TO BECOME AVAILABLE – FREIBURG UNIVERSITY SEEMS TO BE SLOW**. The author will be happy to

support interested readers in reproducing the results and performing subsequent research.

*Author contributions.* N/A

*Competing interests.* The author declares that he has no competing interests.

*Acknowledgements.* The author would like to thank two anonymous reviewers for their thorough consideration for their very constructive suggestions to improve to readability of the manuscript. The author would also like to thank Wolfgang Schwanghart for the editorial handling.

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
