# Peer review of "Rivers as linear elements in landform evolution models"

_Earth Surface Dynamics, 2019_

## Referee Comment (RC1) · Taylor Perron (Referee) · 21 Jan 2020

Dear author,

A point of clarification: Paragraph 23 of Perron et al. (2008) reads,

> The factor $w_{i,j}/\delta$ in equation (15) (where $\delta$ is the grid spacing; in the present study, $\delta = \Delta x = \Delta y$ in all cases), a modification due to Howard (1994a), accounts for the fact that stream channels have a finite width (equation (11)) that is narrower than the grid spacing. Neglecting this factor would assume implicitly that channels have a width $\delta$, and the model solutions would then be resolution-dependent. In this study, we assume for simplicity that $k_4 = 1\,\mathrm{m}$ and $b = 0$ in equation (11) [$w = k_4 Q_w^b$], noting that future studies intending to apply this model to a specific field site will need to

calibrate the channel width function. The values of $k_4$ and $b$ do affect the form of the model topography, but using typical measured values that lead to spatially variable channel width would not qualitatively change the results presented here.

So the analysis in that paper uses an approach similar to that of Howard (1994) and does consider the physical meaning of the channel width, even if the numerical experiments assume it is spatially uniform!

Best regards,

Taylor Perron

---

## Author Comment (AC1) · 21 Jan 2020

Dear Taylor Perron,

thanks for starting the discussion so soon! I think we need to discuss your point in more detail.

I think I got the subpixel approach of Howard's model and the meaning of the channel width $w$ correctly. However, my key point is that all the stream power stuff comes from the analysis of river profiles, starting from the work of Hack. These rivers probably erode their bed plus the adjacent hillslopes (the material coming from the hillslopes into the river). Taking this into account, my result is that the parameter $w$ in Howard's model is not channel width, but a property depending on the threshold where fluvial erosion starts and on the network topology. This property is indeed constant for the entire drainage network, and it scaling properties are developed in my manuscript. So

Howard's (and your) approach is correct with constant $w$, except that $w$ is physically not the channel width. In this sense I would only disagree to your statement "noting that future studies intending to apply this model to a specific field site will need to calibrate the channel width function." Inserting a channel width depending on the catchment size for $w$ and leaving the rest as it would make the model inconsistent with observed river profiles.

Best regards,
Stefan Hergarten

―――――――――――――――――――――

---

## Referee Comment (RC2) · Anonymous Referee #2 · 24 Feb 2020

I applaud Hergarten for taking on an important problem in geomorphic modeling: how to achieve grid-resolution independence in coupled colluvial-fluvial landscape evolution models. All numerical models must be grid-resolution-independent solutions of the underlying equations. That is, they must converge to some validated solution as the pixel size decreases. Many landscape evolution models, however, are not grid-resolution independent and others achieve grid-resolution independence using ad hoc means. This is a significant issue that hinders our ability to model landscape evolution and to compare the results of one model to another.

Before providing my comments on the manuscript, I wish to first review a key alternative approach to the problem as a means of introducing the general issues at play. Pelletier (2010) addressed the problem of grid-resolution dependence in coupled hillslope-channel landscape evolution models by first noting that contributing area scales linearly with pixel size on hillslopes and other areas of non-convergent water

flow but independently of pixel size in fluvial valleys and other areas of convergent water flow. Given that all fluvial erosion formulae are based on unit or specific discharges (i.e., discharges per unit channel width) or related quantities such as shear stress or unit stream power, Pelletier proposed that landscape evolution models should also be based on unit or specific contributing area, both as a means of more faithfully acting as a proxy for unit or specific discharge and for minimizing the general grid-resolution dependence of flow-routing algorithms he documented. With the different widths of water flow on hillslopes (where flow occurs as sheetflow or rill flow) and channels (where flow is confined to a width that may be smaller than a pixel width) taken into account, Pelletier (2010) demonstrated how grid-resolution-independent fluvial incision rates can be computed. It is important to emphasize that any fluvial incision rate must depend, implicitly or explicitly, on the width of water flow since putting more total discharge into a narrower flow pathway will increase the fluvial erosion rate if steepness, climate, and rock characteristics are unchanged.

Pelletier (2010) also demonstrated that it was necessary to scale the divergence of colluvial sediment flux by the ratio of the width of water flow to the pixel width because otherwise colluvial deposition rates (which, together with the fluvial erosion rate, controls the rate of elevation change) in narrow valley bottoms are systematically underpredicted in models. Take, as an example, a location where the valley bottom in nature that the model is trying to represent is only 1 m wide but the pixel width is 10 m. The divergence term will be dominated by the gradient in valley side slopes in the cross-sectional direction. The model will predict that gradient to be the difference in slopes on either side of the valley divided by 10 m, while the actual gradient in nature will be the difference in slopes on either side of the valley divided by 1 m. The colluvial deposition term will, therefore, be too small by a factor of 10 in the model relative to nature and hence must be scaled up accordingly. It is important to note that such a modification to the colluvial deposition term is not some indirect way of scaling the fluvial term as Hergarten implies. Far from being a "problem obviously coming from the fluvial incision term" (line 60), it addresses a limitation of the model to represent the

cross-valley curvature and the effect of that limitation on the resultant model-predicted colluvial deposition rates in valley bottoms.

Grid-resolution dependence in coupled colluvial-fluvial models can be seen most readily as a dependence of drainage density on pixel size. If I understand correctly, Hergarten is proposing to use this variation/error in drainage density to scale the fluvial erosion term. I am wary of this approach because there is no clear (at least to me) physical basis for why the fluvial erosion term would need to be scaled in this way and because there is no indication that the drainage density predicted by the model, even if it can be shown to be grid-resolution independent, is the correct one for a given set of model parameters after such scaling.

I apologize if I missed it, but I didn't see that Hergarten demonstrated that his approach actually leads to grid-resolution-independent results. I was expecting to see model results with similar topography as the pixel size varies over a wide range. No such figure appears in the paper. I recommend that Hergarten present such a figure along with any other analysis (e.g., predicted steady state drainage density as a function of pixel size) needed to demonstrate grid-resolution independence of the model predictions. I would like to see such grid-resolution independence also demonstrated for cases on non-uniform uplift rates, as such applications are common in landscape evolution models.

I had a hard time following the description of the scaling approach. My understanding is that the hillslopes and channels in the model output are first differentiated using a user-defined threshold area, Ac, and then the fluvial erosion term is modified by an amount equal to a power-law function of Ac. The power-law modification to Ac is clear but how is Ac chosen? Does the model have to be run first without scaling the fluvial erosion term in order to determine Ac and then rerun with the scaling? Please provide a step-by-step guide for performing the proposed scaling that is applicable not just to the case of steady uniform uplift to steady state but for other potential landscape evolution model applications. It may be that for the case of steady uniform uplift,

channels and hillslopes can be differentiated based on a threshold contributing area, but many landscape evolution models are of non-uniform uplift and hence non-uniform drainage density. Moreover, there is a large literature on how to differentiate hillslopes and channels both in models and real-world DEMs, and the use of a single contributing area threshold is universally regarded as an inadequate approach to such differentiation. Assuming that choosing Ac involves differentiating hillslopes and channels before scaling the fluvial erosion term, this manuscript glosses over a very complex topic, the implications of which likely influences the applicability of the proposed method.

A minor issue: it is incorrect to state that the erodibility coefficient K depends on rock characteristics and precipitation (line 25). K is influenced by any factor other than channel slope and contributing area that influences detachment-limited erosion rates, including channel width, all of the factors that influence rainfall-runoff partitioning (including vegetation, soil texture, the distribution and sequence of storm events), snowmelt dynamics (for some catchments), etc.

Again, I applaud Hergarten for taking on this important problem and look forward to seeing clarifications in due course.

---

## Author Comment (AC2) · 29 Feb 2020

Dear Reviewer,

thanks for your constructive and encouraging comments! It is good to see that the problem with the grid-resolution dependence is still an important issue as long as we do not disregard the feedback of the hillslopes to the rivers. Let me take the chance to clarify your points immediately, starting from the discussion of the alternative approach suggested by Pelletier (2010).

I agree to your first point in recapitulating Pelletier's (2010) approach – *Given that all fluvial erosion formulae are based on unit or specific discharges (i.e., discharges per unit channel width) or related quantities such as shear stress or unit stream power, Pelletier proposed that landscape evolution models should also be based on unit or specific contributing area, . . . .* They should, and this would make things easier. How-

ever, the relation that is used in almost all models is an erosion rate as a function of channel slope and total catchment size (some proxy for total discharge, but not for discharge per unit width). This old relationship was empirically derived from river profiles and has the advantage that it immediately bridges between model results and real river profiles via the concavity index and the steepness index. It is a lumped equation that already includes the downstream increase of river width without specifying this increase explicitly. This means that we must be careful when starting from this relation and bringing river width into play afterwards. In my opinion, this applies to both the concept suggested by Howard (1994) and adopted by Perron et al. (2008) as well as Pelletier's approach.

My interpretation of Pelletier's idea was slightly different from your explanation. I thought of taking the entire flux from both hillslopes into the river and distributing it not over a width $\delta x$, but over the river width $w$. However, the representation in the equation is the same in both cases, and I guess that you know better than me what the original idea was. The scaling is opposite to the other concept as it rescales the divergence of the hillslope flux at the river by $\frac{\delta x}{w}$ instead of rescaling the fluvial erosion rate by $\frac{w}{\delta x}$. This was the reason for my sloppy argument about the "problem obviously coming from the fluvial incision term."

The main problem, however, is that Pelletier's concept is not independent of the spatial resolution at least according to my preliminary findings. The results of a simple numerical experiment are attached as figures. It describes a river segment of unit length and width (one slope of length $\frac{1}{2}$ at each side of the river). All parameters are set to unity (including the uplift rate and the catchment size in the stream power law) except for a diffusivity of 0.1. Figs. 1–3 show the topography (with the scaling approach suggested in my manuscript) at different times, starting from a flat topography. Figs. 4–6 show the river profile for $\delta x = 0.01$ and $\delta x = 0.1$ at different times with both scaling concepts. The profile should approach a straight line with a slope of 1 for $t \to \infty$. While the profile becomes independent of $\delta x$ for large $t$ for both scaling concepts (although too

steep in Pelletier's approach), the time scale of adjustment strongly depends on $\delta x$ for Pelletier's scaling concept. This means that the time scale of response to changes in uplift etc. depends on the spatial resolution in Pelletier's scaling concept, so that I am not convinced that is solves the scaling issue. However, I may be wrong, and if you ever tested the scaling properties of Pelletier's approach and obtained different results, I would be happy to know.

Now about the specific points addressed in your review.

*Grid-resolution dependence in coupled colluvial-fluvial models can be seen most readily as a dependence of drainage density on pixel size.*

I do not fully agree to this statement. If we assume that rivers start at points with a given minimum catchment size $A_c$ (in m$^2$, not in DEM pixels) and a well-organized dendritic network (not parallel flow on slopes), the dependence of drainage density on DEM resolution is rather weak.

*If I understand correctly, Hergarten is proposing to use this variation/error in drainage density to scale the fluvial erosion term.*

The dominator is indeed something like drainage density except for two differences: (i) Area is not total area as it is in drainage density, but only the part of the area not draining to leaves of the river network. This is the part that makes my analysis a bit complicated at first sight. (ii) Total river length is area of the DEM that covers the network divided by mesh width. For a square grid, this means that diagonal river segments have the same length as those in direction of the axes.

*I am wary of this approach because there is no clear (at least to me) physical basis for why the fluvial erosion term would need to be scaled in this way and because there is no indication that the drainage density predicted by the model, even if it can be shown to be grid-resolution independent, is the correct one for a given set of model parameters after such scaling.*

As mentioned above, I would immediately buy this argument if the widely used model was derived from physical principles. Then the rivers would not know about properties such as drainage density. However, it comes from empirical data of "typical" rivers eroding "typical" landscapes. My conjecture is that the expression for the fluvial erosion law, in particular the value of the erodibility, refers to an equilibrium of erosion and uplift in the catchment and does not describe the river as an isolated object. If this conjecture holds, scaling must be like the one described in my manuscript.

*I apologize if I missed it, but I didn't see that Hergarten demonstrated that his approach actually leads to grid-resolution-independent results. I was expecting to see model results with similar topography as the pixel size varies over a wide range. No such figure appears in the paper. I recommend that Hergarten present such a figure along with any other analysis (e.g., predicted steady state drainage density as a function of pixel size) needed to demonstrate grid-resolution independence of the model predictions. I would like to see such grid-resolution independence also demonstrated for cases on non-uniform uplift rates, as such applications are common in landscape evolution models.*

I am afraid that you did not miss it. I thought it would be clear from the analyzed network properties alone, but accept that it is not. So I can include a more serious version of the simulated river segment and the results of a series of larger simulations that I have just started. In order to make it a bit more interesting, these simulations use irregular triangular grids with $10^5$ to $10^7$ nodes and a threshold model for the hillslopes. I think they will be ready in a few weeks.

*I had a hard time following the description of the scaling approach. My understanding is that the hillslopes and channels in the model output are first differentiated using a user-defined threshold area, Ac, and then the fluvial erosion term is modified by an amount equal to a power-law function of Ac. The power-law modification to Ac is clear but how is Ac chosen? Does the model have to be run first without scaling the fluvial erosion term in order to determine Ac and then rerun with the scaling?*

It is much easier than you think, and the practical relevance of the value of $A_c$ is limited in most applications. The result of my approach is that the erodibility $K$ as it is usually considered is not the parameter that we need, but $K$ multiplied by a length constant (which is not the river width) instead. I suggest $\sqrt{2A_c}$ as a simple estimate of this length scale. If we use a given erodibility $K$, we expect a certain channel steepness in equilibrium with a given uplift rate. The only prediction of my concept is that we can define any value $A_c$ and let fluvial erosion act only at catchment sizes $A > A_c$, we will arrive at the correct channel steepness. In many applications there will be hillslope processes affecting scales larger than $A_c$. If these are strong, fluvial erosion will lose relevance even for for $A > A_c$, and the value of $A_c$ also becomes less relevant. If it is much smaller than the scale of the hillslope process, it even only defines the reference topography that would occur if the considered hillslope process was switched off.

*Please provide a step-by-step guide for performing the proposed scaling that is applicable not just to the case of steady uniform uplift to steady state but for other potential landscape evolution model applications. It may be that for the case of steady uniform uplift, channels and hillslopes can be differentiated based on a threshold contributing area, but many landscape evolution models are of non-uniform uplift and hence non-uniform drainage density. Moreover, there is a large literature on how to differentiate hillslopes and channels both in models and real-world DEMs, and the use of a single contributing area threshold is universally regarded as an inadequate approach to such differentiation. Assuming that choosing Ac involves differentiating hillslopes and channels before scaling the fluvial erosion term, this manuscript glosses over a very complex topic, the implications of which likely influences the applicability of the proposed method.*

Not really – it is all only about bringing empirically determined values of $K$ into the model. We are free to assume any model for fluvial erosion at small scales such as a spatially variable threshold or a continuous decrease of erosion rates at decreasing catchment sizes. We just have to keep in mind that the value of $K$ is the one that

we would measure from equilibrium river profiles if we assume that fluvial erosion is switched on for $A > A_c$. In this approach, differentiating hillslopes from channels on a given topography would only be useful if we want to use a specific value of $K$ measured in a given catchment. If we knew the spatial distribution of erosion in this catchment, we could use it for assigning a "realistic" value of $A_c$ to this value of $K$. However, this is hopeless in most cases, so that we have to accept the problem that measured values of $K$ many unresolved dependencies (including something like $A_c$) as you already mentioned.

*A minor issue: it is incorrect to state that the erodibility coefficient K depends on rock characteristics and precipitation (line 25). K is influenced by any factor other than channel slope and contributing area that influences detachment-limited erosion rates, including channel width, all of the factors that influence rainfall-runoff partitioning (including vegetation, soil texture, the distribution and sequence of storm events), snowmelt dynamics (for some catchments), etc.*

Finally, at least one point where I agree without any reservations.

Best regards,
Stefan Hergarten

[Figure]

[Figure]

**t = 0.03**

[Figure]

**Fig. 1.**

**t = 0.3**

(3D surface plot with axes labeled H (vertical, 0 to 0.3), x (0 to 1), and y (-0.5 to 0.5))

**Fig. 2.**

**t = 3**

H

y

x

**Fig. 3.**

**Pelletier (2010) scaling**

**Legend:**
- t = 0.03, δx = 0.01
- t = 0.1, δx = 0.01
- t = 0.3, δx = 0.01
- t = 1, δx = 0.01
- t = 3, δx = 0.01
- t = 10, δx = 0.01
- t = 0.03, δx = 0.1
- t = 0.1, δx = 0.1
- t = 0.3, δx = 0.1
- t = 1, δx = 0.1
- t = 3, δx = 0.1
- t = 10, δx = 0.1

**Fig. 4.**

**Pelletier (2010) scaling**

Legend:
- t = 0.03, δx = 0.01
- t = 0.1, δx = 0.01
- t = 0.3, δx = 0.01
- t = 1, δx = 0.01
- t = 3, δx = 0.01
- t = 10, δx = 0.01
- t = 0.03, δx = 0.1
- t = 0.1, δx = 0.1
- t = 0.3, δx = 0.1
- t = 1, δx = 0.1
- t = 3, δx = 0.1
- t = 10, δx = 0.1

**Fig. 5.**

**Scaling suggested here**

- t = 0.03, δx = 0.01
- t = 0.1, δx = 0.01
- t = 0.3, δx = 0.01
- t = 1, δx = 0.01
- t = 3, δx = 0.01
- t = 10, δx = 0.01
- t = 0.03, δx = 0.1
- t = 0.1, δx = 0.1
- t = 0.3, δx = 0.1
- t = 1, δx = 0.1
- t = 3, δx = 0.1
- t = 10, δx = 0.1

**Fig. 6.**

---

## Referee Comment (RC3) · Anonymous Referee #3 · 19 Mar 2020

Overview:

This paper addresses an important, and still largely unsolved, question in landscape evolution modeling: how to eliminate or minimize grid-scale dependence that can arise from the combination of flow algorithm, water erosion law, and gravitational transport law. The paper makes the case for a new approach. In my opinion, this is a valuable contribution that should be published, after suitable modifications. I have some recommendations, below, for how the value and impact of the paper could be increased. The most important of these is a recommendation to show examples of model simulations with and without the proposed scaling solution, to demonstrate that the solution does indeed minimize mesh-size dependence (and also to illustrate the nature of the problem generally).

General comments:

[Figure]

(1) The scaling problem: the paper describes in words the scaling problem as it manifests in landscape evolution models, but a picture would be worth a thousand words. I think the impact of the paper would be greater if the author added a figure showing visually the effect discussed section 2: the steepening of topography with decreasing pixel size. Pelletier (2010) has a figure showing the (relative) lack of such effects when using his proposed solution, so one idea would be to mirror that figure (his Fig 6) but without any attempt to scale the problem away. You could even use the same parameters. I would also suggest including a plot showing equilibrium slope-area scaling for models at different pixel resolutions.

Then, follow up by showing the same examples, but now using the proposed scaling solution. This would (presumably) demonstrate that the solution works. Adding such a 'before and after' pair of figures seems really key to selling the core idea of the paper; otherwise, readers might be left wondering 'if I bother to do this, will it really work?'.

(2) Some parts of the paper come across as if written for readers who are already well familiar with the relevant literature. I recommend making a few small additions / modifications that would make the paper accessible to, for example, graduate students who are just starting out, or people in other fields who have a new interest in the topic. Mostly this is a matter of adding example references to the literature and/or expanding on some points, as noted in the specific comments below.

(3) A general question, which would be worth answering somewhere in the text, is whether the scaling analysis still holds if the drainage patterns are qualitatively different on hillslopes versus channels. In this Discussion version of the paper, it is not clear whether the simulation in Figure 1 includes any local transport (ie hillslope) processes; if so, it is not apparent in the drainage patterns.

Notes keyed to text by line number:

19-20 For readers unfamiliar with this idea, it would be helpful to add one or a few example references (one of the earliest I am aware of is Andrews and Bucknam, 1987;

another option would be to cite a review paper that discusses mathematical representations of various geomorphic processes)

23 'has become some kind of paradigm' - this will not mean much to readers who are just joining the conversation. Suggest giving one or a few example references.

24 I disagree that equation 2 (I think that is what is meant by 'it') requires the assumption of constant precipitation. First, I would argue that it is runoff rather than precipitation per se that matters (though obviously the two are correlated). Second, there are quite a few papers that discuss the relation between precipitation characteristics and formulas like equation 2, going back at least to Willgoose et al. (1991 'part 1' in Water Resources Research, their Appendix B), and continuing more recently with papers like Deal et al. (2018) (and lots of literature in between). In any event, the text about 'constant precipitation' (constant in space or time or both?) seems like just a side comment, and maybe the best approach would be simply to delete it.

26 As above, I would argue it is the spatial distribution of runoff that matters most; precipitation has some influence on this, but there are other factors too.

33 The relation predicts m/n = theta ONLY if the erosion rate and erodibility are uniform in space and steady in time. You allude to that in the next sentence, but the way this is worded would be confusing for a reader who does not understand that you are referring to a special case here. I recommend re-wording this section to be more precise.

35-38 I would argue that the condition of equilibrium is more general than the word 'uplift' implies. The key is that the erosion rate is space-time uniform. This could be due to actual tectonic uplift relative to, say, sea level. Or it could be an equilibrium relative to a given rate of base-level lowering at the boundary of a given system (and in fact the former is a subset of the latter).

42-43 'the total area covered by large rivers decreases with decreasing mesh width': can you provide evidence for this, or otherwise clarify this concept? It seems to disagree with the view of Pelletier (2010), who showed examples where the drainage area of the larger catchments remains invariant to grid resolution, whereas the drainage area associated with hillslope 'patches' decreases with pixel size. My sense is that the area of the larger catchments in a given DEM or landscape model is probably strongly influenced by the domain size and geometry. Maybe what you actually mean here is that the surface area covered by stream segments ('channel pixels'), rather than drainage area, shrinks as pixel size shrinks (tending toward zero when the network segments become infinitesimally wide linear features).

46 - I think there is a bit more to it than that. If you omit local transport (ie, diffusion or diffusion-like modification of the topography), you have the odd circumstance where for the equilibrium case the equations predict that H => infinity as A => 0. In practical terms, then, a model with just eq 2 would have increasing relief with decreasing pixel size. Might be worth pointing out, as the current text ('scaling problem may not be critical') could be misinterpreted as meaning there is no pixel size dependence without local transport.

58-61 I don't think this summary quite does justice to Pelletier (2010). I suggest adding something like 'where the factor is unity on cells identified as hillslopes, but greater than unity for cells that represent valley features'. Also, you might add that a reason to suspect it doesn't work for 'all types of local transport' is that his derivation was (heuristically at least) based on a linear model.

62 Can you expand here to say why large mesh widths would be immune? Large relative to what? Is the idea that if all cells are conceptually valley cells, then you don't need special treatment for hillslopes versus valleys?

75 For what it's worth, I would argue that 'bedrock incision' just means what it says, and does not (or at least should not) imply any particular mechanism or model thereof. I think the idea you are trying to get at here is that there is a difference between assuming that a channel must entrain and remove only the material on the channel bed,

or that it must entrain and remove that plus the sum of material transported into the channel from surrounding hillslopes. I do not think the term 'bedrock erosion' is all that helpful in articulating the difference between these two possibilities, but it would be worth expanding on the idea: for example to note that it depends on the degree of contrast between the 'mobile' material coming from side slopes and the 'intact' or 'original' material in the channel floor (one example of highly resistant material coming from side slopes is Shobe et al. (2016 GRL)).

86-88 Consider noting here that Pelletier (2010) described an alternative approach based on comparing computing drainage area on the DEM grid, and on a 2x higher resolution interpolated version of the DEM. That approach has the advantage of allowing the processes to determine the drainage density. I suspect that the mechanism for identifying channel versus hillslope pixels probably does not matter much for the technique you propose, and if that is the case, then it would be worth pointing out. For the sake of developing the idea, using a fixed Ac seems totally fine. But as a reader I would like to know whether I can still use the approach if I use a different method for distinguishing channel and hillslope pixels.

89-90 I got confused at first by the definition of Ae. A key aspect of the definition is that it includes only those pixels that drain DIRECTLY to a given channel pixel, and not ones that 'pass through' another channel pixel upstream. If that understanding is correct, it would be worth stating this (because other readers, like me, are probably used to thinking of contributing area as something that accumulates downstream).

Eq 5: unless you are changing the definition of K, this equation seems to change the meaning of E: in eq 2 it seems to be length per time, but in eq 5 it seems to become volume per time. If that is correct, I recommend using a different symbol than E to avoid confusion. Note I am assuming that Ae is a surface area. The text says 'number of sites', so I guess it is actually meant to be dimensionless (just a count), as text later in the paper implies. But in that case then you're no longer talking about a physical law. Why not treat Ae as a surface area, and either have the equation represent the

volumetric erosion rate over the area concerned, or divide by cell area to arrive at a length per time. At any rate, clarification of these issues in the text would be helpful.

94 - It would be very helpful to add more information about this model and the conditions under which it was run to generate figure 1. Is OpenLEM in this example solving just eq 2 or does it include diffusion too? What flow routing algorithm does it use? How are closed depressions handled? Was it run until steady state balance between erosion and uplift/baselevel was reached? Does the fluvial threshold Ac actually apply in the numerical model, ie, are areas smaller than Ac treated exclusively with local transport? Is local transport applied to all pixels or just those A < Ac? Or, alternatively, was the model run without any threshold or hillslopes? In addition, please list all the input parameters so readers could reproduce or replicate the experiment.

121 I think you mean 'site' not 'size'

120-125 and eq 6: I found this section confusing. I understand Ae to be a spatial field, with a different value at every pixel. Yet if P(Ac) is just a scalar fraction, then eq 6 implies a unitary value for Ae. Is your aim here a derived distribution of the cumulative probability of Ae? ... Ah ok, reading later, you mention Ae is dimensionless (but perhaps you can see why it is confusing given that A and Ac refer to areas).

134 I recommend a more extensive explanation here. Clearly figure 4 shows that the Ae-Ac relation follows a power law with about the same slope as that of the cumulative area distribution. But the underlying scaling argument is hard to follow.

151 It would be helpful to know the parameters used to generate these synthetic topographies.

eq 9: if you used this directly in a model, would it not break equilibrium slope-area scaling? Or are you suggesting that the leading factors compensate for deposition by material sourced from surrounding hillslopes? I can imagine such an argument being quantified as:

equilibrium => fluvial erosion rate = uplift (baselevel) rate + hillslope deposition rate

=> E = U + D

and the deposition rate is (Ae - 1) U, ie deposition from the hillslope area but not the pixel itself, so you have

E = Ae U

...etc. This is basically the argument you're making, right? That effectively a fluvial grid cell has to drill through not only its own material, but also all the material coming from the surrounding hills. I think the idea would be conveyed more clearly if you added some math along the lines of the above.

183-4 reference for this number?

185-eq 10: I can see the advantage of this approach, but would like to see some discussion of how to reconcile the concept of a threshold area Ac with the actually valley head area that emerges from a model. To mirror my questions above, are you suggesting that this approach should be paired with using a model that only applies fluvial erosion to locations with A > Ac, where Ac is a parameter? Or could one allow Ac to emerge from the dynamics, as in Pelletier (2010)?

205-6 good point, and some models I'm aware of allow for diffusion-like transport to be applied ONLY to convex locations, with the assumption that the material is instantly carried away in concave-up locations.

Code availability: I do not know what the policy of Esurf is, but 'available on request' is no longer generally considered best practice. Better to place code in a community repository, or at least a public repository. Better still to have it under version control. Even better yet to provide input files, examples of usage, etc., in an open repository (see Wilson et al. below).

Some example literature on open and reproducible research software:

Nick Barnes. Publish your computer code: it is good enough. Nature, 467(7317): 753?753, 2010.

Irving, D. (2016). A minimum standard for publishing computational results in the weather and climate sciences. Bulletin of the American Meteorological Society, 97(7), 1149-1158.

Wilson, G., Bryan, J., Cranston, K., Kitzes, J., Nederbragt, L., & Teal, T. K. (2017). Good enough practices in scientific computing. PLoS computational biology, 13(6).

Benureau, F. C., & Rougier, N. P. (2018). Re-run, repeat, reproduce, reuse, replicate: transforming code into scientific contributions. Frontiers in neuroinformatics, 11, 69.

Stodden, V., Krafczyk, M. S., & Bhaskar, A. (2018, June). Enabling the verification of computational results: An empirical evaluation of computational reproducibility. In Proceedings of the First International Workshop on Practical Reproducible Evaluation of Computer Systems (pp. 1-5).

Chen, X., Dallmeier-Tiessen, S., Dasler, R., Feger, S., Fokianos, P., Gonzalez, J. B., ... & Rodriguez, D. R. (2019). Open is not enough. Nature Physics, 15(2), 113-119.

---

## Author Response (AR1)

Dear Reviewers,

thanks for your constructive and encouraging comments, in particular to the two anonymous reviewers who made many suggestions to improve the accessibility of the manuscript to a wider readership. It is good to see that the problem with the grid-resolution dependence is still an important issue. In the following, the points addressed in your reports are discussed, and changes to the manuscript are described. Line numbers refer to the version with highlighted changes.

**Reviewer 1**

"A point of clarification: Paragraph 23 of Perron et al. (2008) reads, ... So the analysis in that paper uses an approach similar to that of Howard (1994) and does consider the physical meaning of the channel width, even if the numerical experiments assume it is spatially uniform!"

**Reviewer 2**

"Before providing my comments on the manuscript, I wish to first review a key alternative approach to the problem as a means of introducing the general issues at play. Pelletier (2010) addressed the problem of gridresolution dependence in coupled hillslopechannel landscape evolution models ... It is important to note that such a modification to the colluvial deposition term is not some indirect way of scaling the fluvial term as Hergarten implies. Far from being a "problem obviously coming from the fluvial incision term" (line 60), it addresses a limitation of the model to represent the cross-valley curvature and the effect of that colluvial deposition rates in valley bottoms."

I think I got the subpixel approach of Howard's model and the meaning of the channel width w correctly. However, my key point is that your version with constant channel width is in principle correct, but somehow for the wrong reason as the length scale that is needed for compensating the mesh width  $\delta$  is not the channel width, but another property related to the threshold catchment size where fluvial erosion starts, and that this property is indeed constant over the drainage network. I explained this point more clearly in the revised version by extending the description of the problem (Sect. 2) and the discussion (Sect. 5).

I must admit that my discussion of Pelletier's (2010) approach was way too short. I do not want to raise any doubts against the major part of this paper addressing flow routing and distinguishing between channelized flow and parallel flow. However, I am not convinced by the scaling approach itself, i.e., rescaling the divergence of the flux from the hillslopes. If my understanding and my own analysis are not completely wrong, this approach suffers from the same problem as Howard/Perron version. As soon as river width increases with catchment size, river steepness is no longer consistent with the empirical findings of Hack (1957) unless the exponent m is changed. In this case, however, the relationship to the widely used concept of the erodibility is lost. According to my findings, this problem affects both approaches suggested previously in almost the same way. The numerical example with the parallel rivers given by Pelletier (2010) navigated around this problem by considering relief and valley spacing. I discussed the problem more thoroughly in Sect. 2 now (lines 86-114).

"Grid-resolution dependence in coupled colluvial-fluvial models can be seen most readily as a dependence of drainage density on pixel size."

"If I understand correctly, Hergarten is proposing to use this variation/error in drainage density to scale the fluvial erosion term."

"I am wary of this approach because there is no clear (at least to me) physical basis for why the fluvial erosion term would need to be scaled in this way and because there is no indication that the drainage density predicted by the model, even if it can be shown to be grid-resolution independent, is the correct one for a given set of model parameters after such scaling."

"I apologize if I missed it, but I didn't see that Hergarten demonstrated that his approach actually leads to grid-resolutionindependent results. I was expecting to see model results with similar topography as the pixel size varies over a wide range. No such figure appears in the paper. I recommend that Hergarten present such a figure along with any other analysis (e.g., predicted steady state drainage density as a function of pixel size) needed to demonstrate gridresolution independence of the model predictions. I would like to see such grid-resolution independence also demonstrated for cases on non-uniform uplift rates, as such applications are common in landscape evolution models."

I do not fully agree to this statement. If we assume that rivers start at points with a given minimum catchment size  $A_c$  (in m2, not in DEM pixels) and a well-organized dendritic network (not parallel flow on slopes), the dependence of drainage density on DEM resolution is rather weak. It is rather the total area covered by the DEM pixels. This should be clearer now in the **more detailed explanation (lines 153–171).**

The dominator is indeed something like drainage density except for two differences: (i) Area is not total area as it is in drainage density, but only the part of the area not draining to leaves of the river network. This is the part that makes my analysis a bit complicated at first sight. (ii) Total river length is area of the DEM that covers the network divided by mesh width. For a square grid, this means that diagonal river segments have the same length as those in direction of the axes. This should also be clearer with the **new explanation (lines 153–171).**

I would immediately buy this argument if the widely used model was derived from physical principles. Then the rivers would not know about properties such as drainage density. However, it comes from empirical data of "typical" rivers eroding "typical" landscapes. My conjecture is that the expression for the fluvial erosion law, in particular the value of the erodibility, refers to an equilibrium of erosion and uplift in the catchment and does not describe the river as an isolated object.

I am afraid that you did not miss it. I only thought about the theoretical concept and the generic hillslope process model that follows the direction of the river pattern where it is somehow clear that it should work. You are right, it is not so clear, in particular if we use a "realistic" hillslope process model. I have now added a new section (Sect. 4) with 2 numerical examples of different complexity. "I had a hard time following the description of the scaling approach. My understanding is that the hillslopes and channels in the model output are first differentiated using a userdefined threshold area,  $A_c$ , and then the fluvial erosion term is modified by an amount equal to a power-law function of  $A_c$ . The power-law modification to  $A_c$  is clear but how is  $A_c$  chosen? Does the model have to be run first without scaling the fluvial erosion term in order to determine  $A_c$  and then rerun with the scaling?"

"Please provide a step-by-step guide for performing the proposed scaling that is applicable not just to the case of steady uniform uplift to steady state but for other potential landscape evolution model applications. It may be that for the case of steady uniform uplift, channels and hillslopes can be differentiated based on a threshold contributing area, but many landscape evolution models are of non-uniform uplift and hence nonuniform drainage density. Moreover, there is a large literature on how to differentiate hillslopes and channels both in models and real-world DEMs, and the use of a single contributing area threshold is universally regarded as an inadequate approach to such differentiation. Assuming that choosing  $A_c$  involves differentiating hillslopes and channels before scaling the fluvial erosion term, this manuscript glosses over a very complex topic, the implications of which likely influences the applicability of the proposed method."

Sorry for this! It is much easier than you think, and the practical relevance of the value of  $A_c$  is limited in most applications. The result of my approach is that the erodibility K as it is usually considered is not the parameter that we need, but K multiplied by a length constant (which is not the river width) instead. I suggest  $\sqrt{2A_c}$  as a simple estimate of this length scale. If we use a given erodibility K, we expect a certain channel steepness in equilibrium with a given uplift rate. The only prediction of my concept is that we can define any value  $A_c$  and let fluvial erosion act only at catchment sizes  $A > A_c$ , we will arrive at the correct channel steepness. In many applications there will be hillslope processes affecting scales larger than  $A_c$ . If these are strong, fluvial erosion will lose relevance even for for  $A > A_c$ , and the value of  $A_c$  also becomes less relevant. If it is much smaller than the scale of the hillslope process, it even only defines the reference topography that would occur if the considered hillslope process was switched off. I hope that this has become clearer in the revised version, in particular with the help of the numerical example in Sect 4.

Not really – it is all only about bringing empirically determined values of K into the model. We are free to assume any model for fluvial erosion at small scales such as a spatially variable threshold or a continuous decrease of erosion rates at decreasing catchment sizes. We just have to keep in mind that the value of K is the one that we would measure from equilibrium river profiles if we assume that fluvial erosion is switched on for  $A > A_c$ . In this approach, differentiating hillslopes from channels on a given topography would only be useful if we want to use a specific value of K measured in a given catchment. If we knew the spatial distribution of erosion in this catchment, we could use it for assigning a "realistic" value of  $A_c$  to this value of K. However, this is hopeless in most cases, so that we have to accept the problem that measured values of K many unresolved dependencies (including something like  $A_c$ ) as you already mentioned.

"A minor issue: it is incorrect to state that the erodibility coefficient K depends on rock characteristics and precipitation (line 25). K is influenced by any factor other than channel slope and contributing area that influences detachment-limited erosion rates, including channel width, all of the factors that influence rainfall-runoff partitioning (including vegetation, soil texture, the distribution and sequence of storm events), snowmelt dynamics (for some catchments), etc."

**Reviewer 3**

"(1) The scaling problem: the paper describes in words the scaling problem as it manifests in landscape evolution models, but a picture would be worth a thousand words. I think the impact of the paper would be greater if the author added a figure showing visually the effect discussed section 2: the steepening of topography with decreasing pixel size. Pelletier (2010) has a figure showing the (relative) lack of such effects when using his proposed solution, so one idea would be to mirror that figure (his Fig 6) but without any attempt to scale the problem away. You could even use the same parameters. I would also suggest including a plot showing equilibrium slope-area scaling for models at different pixel resolutions."

"Then, follow up by showing the same examples, but now using the proposed scaling solution. This would (presumably) demonstrate that the solution works. Adding such a 'before and after' pair of figures seems really key to selling the core idea of the paper; otherwise, readers might be left wondering 'if I bother to do this, will it really work?'." Finally, at least one point where I agree without any reservations. I have streamlined the wording (lines 27–29).

I must admit that my explanation of the problem was way too short and written for readers who are already familiar with the problem. I have now extended Sect. 2 and introduced 3 new figures. The new Figure 3 gives a very simple example of the problem. This scenario is used in the following to explain why both approaches suggested before do not solve the problem completely and is also the basis for the first numerical example presented in the new Sect. 4.

I have now added a new section (Sect. 4) with two numerical examples. The first one continues the simple example from Sect. 2 and shows that the approach works perfectly here not only for the steady-state solution, but also concerning the time scale. The second example combines diffusion with fluvial incision and can be seen as some kind of standard scenario for coupling fluvial and hillslope processes. Here the main result is that the approach reduces the scaling problem considerably, but drainage reorganization by hillslope processes leaves a part of the scaling problem.

"(2) Some parts of the paper come across as if written for readers who are already well familiar with the relevant literature. I recommend making a few small additions / modifications that would make the paper accessible to, for example, graduate students who are just starting out, or people in other fields who have a new interest in the topic. Mostly this is a matter of adding example references to the literature and/or expanding on some points, as noted in the specific comments below.'

"(3) A general question, which would be worth answering somewhere in the text, is whether the scaling analysis still holds if the drainage patterns are qualitatively different on hillslopes versus channels. In this Discussion version of the paper, it is not clear whether the simulation in Figure 1 includes any local transport (ie hillslope) processes; if so, it is not apparent in the drainage patterns."

"19-20 For readers unfamiliar with this idea, it would be helpful to add one or a few example references (one of the earliest I am aware of is Andrews and Bucknam, 1987; another option would be to cite a review paper that discusses mathematical representations of various geomorphic processes)"

"23 'has become some kind of paradigm' this will not mean much to readers who are just joining the conversation. Suggest giving one or a few example references."

"24 I disagree that equation 2 (I think that is what is meant by 'it') requires the assumption of constant precipitation. ... In any event, the text about 'constant precipitation' (constant in space or time or both?) seems like just a side comment, and maybe the best approach would be simply to delete it." I tried to make it accessible to a wider readership now – for details see below.

The entire framework was indeed developed for the situation that the fluvial drainage pattern persist on the hillslope as if fluvial erosion was the only erosion process. This also applies to Fig. 1. I tried to point this out more clearly in the revised version and hope the second numerical example in Sect. 4 shows where the remaining problems are.

Andrews and Bucknam (1987) is also the earliest reference to this that I know, although its relationship to coupled models is not really close. I have added this one together with those that I often use as key references in this context (line 21).

Here I would not fully agree. I am quite sure that even most of the readers just joining the conversation about combining fluvial and hillslope processes have either read a at least one paper about modeling fluvial erosion or at least one where the erodibility as a lumped parameter is discussed. In both cases, the chance is quite high that these readers have already seen Eq. (2) or be able to find a paper where it is explained in detail.

Indeed only a side comment, although I would not expect any reader to run into problems with this statement. I have removed it (lines 24–25). "26 As above, I would argue it is the spatial distribution of runoff that matters most; precipitation has some influence on this, but there are other factors too."

"33 The relation predicts m/n = theta ONLY if the erosion rate and erodibility are uniform in space and steady in time. You allude to that in the next sentence, but the way this is worded would be confusing for a reader who does not understand that you are referring to a special case here. I recommend re-wording this section to be more precise."

"35-38 I would argue that the condition of equilibrium is more general than the word 'uplift' implies. The key is that the erosion rate is space-time uniform. This could be due to actual tectonic uplift relative to, say, sea level. Or it could be an equilibrium relative to a given rate of base-level lowering at the boundary of a given system (and in fact the former is a subset of the latter)."

"42-43 'the total area covered by large rivers decreases with decreasing mesh width': can you provide evidence for this, or otherwise clarify this concept? ... Maybe what you actually mean here is that the surface area covered by stream segments ('channel pixels'), rather than drainage area, shrinks as pixel size shrinks (tending toward zero when the network segments become infinitesimally wide linear features)."

"46 - I think there is a bit more to it than that. If you omit local transport (ie, diffusion or diffusion-like modification of the topography), you have the odd circumstance where for the equilibrium case the equations predict that  $H \to \infty$  as  $A \to 0$ . In practical terms, then, a model with just eq 2 would have increasing relief with decreasing pixel size. Might be worth pointing out, as the current text ('scaling problem may not be critical') could be misinterpreted as meaning there is no pixel size dependence without local transport." This is, of course, all true. Originally I just wanted to point out that the erodibility is not only a property of the rock as the term could suggest, but a lumped parameter. I stream-lined this point (lines 27–29).

The sentence should not imply that rivers follow Eq. (3) locally then, but only that the empirical finding of Eq. (3) in many rivers constrains the ratio  $\frac{m}{n}$ . I clarified it in order to avoid any confusion (lines 36– 37).

Of course, but if we replace U with E, Eq. (4) is not even restricted to spatially uniform conditions, but just Eqs. (2) and (3) combined. However, I have replaced U with E in Eq. (4) now.

Yes, of course! Maybe the problem is that I worked with concepts such as box counting in the context of fractals too long, so that I could not imagine that anyone could misunderstand this point. I hope the **rephrased version is clearer (lines 46–47)**.

The revised version describes the scaling properties of the version without local transport in more detail, including two new figures. I would, however, prefer to get around the question whether  $H \to \infty$ for  $A \to 0$ .  $S \to \infty$  is clear, but if we use Hack's (1957) scaling relation between A and upstream length, H remains finite. "58-61 I don't think this summary quite does justice to Pelletier (2010). I suggest adding something like 'where the factor is unity on cells identified as hillslopes, but greater than unity for cells that represent valley features'. Also, you might add that a reason to suspect it doesn't work for 'all types of local transport' is that his derivation was (heuristically at least) based on a linear model."

"62 Can you expand here to say why large mesh widths would be immune? Large relative to what? Is the idea that if all cells are conceptually valley cells, then you don't need special treatment for hillslopes versus valleys?"

"75 For what it's worth, I would argue that 'bedrock incision' just means what it says, and does not (or at least should not) imply any particular mechanism or model thereof. I think the idea you are trying to get at here is that there is a difference between assuming that a channel must entrain and remove only the material on the channel bed, or that it must entrain and remove that plus the sum of material transported into the channel from surrounding hillslopes. I do not think the term 'bedrock erosion' is all that helpful in articulating the difference between these two possibilities, but it would be worth expanding on the idea: for example to note that it depends on the degree of contrast between the 'mobile' material coming from side slopes and the 'intact' or 'original' material in the channel floor (one example of highly resistant material coming from side slopes is Shobe et al. (2016 GRL))."

Yes, the discussion of the approaches by Howard/Perron et al. and Pelletier was indeed much too short. I have now added a more thorough discussion of the properties of both ideas and, in particular, why Pelletier's approach does not solve the problem completely (Sect. 2).

My impression is that it is practically like this. In some studies (including some own), the mesh width is large enough to assume that all sites are channel sites, and hillslope processes are just a small add-on to make the topography more realistic. If we then do not compare simulations with different resolutions and do not mind whether the channel steepness is as expected, it is tempting to disregard the problem. The revised version addresses this aspect more precisely (lines 115–124).

As far as I can see, the term 'bedrock erosion' did not occur; it would indeed be confusing. I must admit that I did not get the point 'bedrock incision just means what it says'. For me it somehow implies that bedrock at the location of the river is eroded, and this is not a particular mechanism or model for me. I still think that the terms 'bedrock incision' and 'detachment-limited erosion' reflect the differences between the two concepts quite well. Nevertheless I agree that including the reference Shobe et al. (2016) is a good idea (lines 133–135).

"86-88 Consider noting here that Pelletier (2010) described an alternative approach based on comparing computing drainage area on the DEM grid, and on a 2x higher resolution interpolated version of the DEM. That approach has the advantage of allowing the processes to determine the drainage density. I suspect that the mechanism for identifying channel versus hillslope pixels probably does not matter much for the technique you propose, and if that is the case, then it would be worth pointing out. For the sake of developing the idea, using a fixed  $A_c$  seems totally fine. But as a reader would like to know whether I can still use the approach if I use a different method for distinguishing channel and hillslope pixels."

"89-90 I got confused at first by the definition of  $A_e$ . A key aspect of the definition is that it includes only those pixels that drain DIRECTLY to a given channel pixel, and not ones that 'pass through' another channel pixel upstream. If that understanding is correct, it would be worth stating this (because other readers, like me, are probably used to thinking of contributing area as something that accumulates downstream). Eq 5: unless you are changing the definition of K, this equation seems to change the meaning of E: in eq 2 it seems to be length per time, but in eq 5 it seems to become volume per time. If that is correct, I recommend using a different symbol than E to avoid confusion. Note I am assuming that  $A_e$  is a surface area. The text says 'number of sites', so I guess it is actually meant to be dimensionless (just a count), as text later in the paper implies. But in that case then you're no longer talking about a physical law. Why not treat  $A_e$  as a surface area, and either have the equation represent the volumetric erosion rate over the area concerned, or divide by cell area to arrive at a length per time. At any rate, clarification of these issues in the text would be helpful."

This is indeed a central point and probably the main limitation of the concept. I hope that this limitation **becomes clearer in the discussion section (Sect. 5) now.**

Ok, I remember that you said that a picture would be worth a thousand words and hope that my more detailed explanation helps in combination with the new Fig. 4. I also hope that it is clearer now that all areas are measured in DEM pixels throughout Sect. 3.

"94 - It would be very helpful to add more information about this model and the conditions under which it was run to generate figure 1. Is OpenLEM in this example solving just eq 2 or does it include diffusion too? What flow routing algorithm does it use? How are closed depressions handled? Was it run until steady state balance between erosion and uplift/baselevel was reached? Does the fluvial threshold  $A_c$  actually apply in the numerical model, ie, are areas smaller than  $A_c$  treated exclusively with local transport? Is local transport applied to all pixels or just those  $A < A_c$ ? Or, alternatively, was the model run without any threshold or hillslopes? In addition, please list all the input parameters so readers could reproduce or replicate the experiment."

"121 I think you mean 'site' not 'size'"

"120-125 and eq 6: I found this section confusing. I understand  $A_e$  to be a spatial field, with a different value at every pixel. Yet if  $P(A_c)$  is just a scalar fraction, then eq 6 implies a unitary value for  $A_e$ . Is your aim here a derived distribution of the cumulative probability of  $A_e$ ? ... Ah ok, reading later, you mention  $A_e$  is dimensionless (but perhaps you can see why it is confusing given that A and  $A_c$  refer to areas)."

"134 I recommend a more extensive explanation here. Clearly figure 4 shows that the  $A_{e}$ - $A_{c}$  relation follows a power law with about the same slope as that of the cumulative area distribution. But the underlying scaling argument is hard to follow."

"151 It would be helpful to know the parameters used to generate these synthetic topographies." Ok, I have now included this information, but quite at the beginning of the paper as the new Fig. 1 already uses topographies of the same type. Except for filling local depressions. These are considered as lakes (deepest outlet) for computing the flow pattern, and erosion is switched off as long as the water level is higher than to topography. However, this is relevant only in the very beginning of the simulations as these lakes vanish soon. I would therefore prefer not to confuse the readers with this additional information.

**Thanks!**

I hope that **this is clearer now.**  $A_e$  defined as the mean size of the contributing areas (in pixels) over all channel sites with a given catchment size. Then the main point is that it is almost independent of the considered catchment size and only depends on  $A_c$ . The confusion with the areas measured in pixels should hopefully dissolve now, too.

I would say it is an immediate consequence of Eq. 12 (numbering of revised manuscript). the argument why it is not exactly the same may indeed be more complicated, but I think it is not a problem for the following parts of the manuscript if it is not immediately understood.

I thought the readers would guess that everything remains the same except for those parameter values explicitly mentioned in Table 1. But in order to state it clearly, **I added a sentence (line 237).**

"eq 9: if you used this directly in a model, would it not break equilibrium slope-area scaling? Or are you suggesting that the leading factors compensate for deposition by material sourced from surrounding hillslopes? ... This is basically the argument you're making, right? That effectively a fluvial grid cell has to drill through not only its own material, but also all the material coming from the surrounding hills. I think the idea would be conveyed more clearly if you added some math along the lines of the above."

"183-4 reference for this number?"

"185-eq 10: I can see the advantage of this approach, but would like to see some discussion of how to reconcile the concept of a threshold area  $A_c$  with the actually valley head area that emerges from a model. To mirror my questions above, are you suggesting that this approach should be paired with using a model that only applies fluvial erosion to locations with  $A > A_c$ , where  $A_c$  is a parameter? Or could one allow  $A_c$  to emerge from the dynamics, as in Pelletier (2010)?"

"205-6 good point, and some models I'm aware of allow for diffusion-like transport to be applied ONLY to convex locations, with the assumption that the material is instantly carried away in concave-up locations."

"Code availability: I do not know what the policy of Esurf is, but 'available on request' is no longer generally considered best practice. Better to place code in a community repository, or at least a public repository. Better still to have it under version control. Even better yet to provide input files, examples of usage, etc., in an open repository (see Wilson et al. below). ...." Yes, exactly like this! I hope that this point becomes clearer with the more detailed explanations in the previous parts, so that there should be no need to extend it here.

A good chance to promote my own paper (line 277).

Yes, indeed with  $A_c$  as a parameter. Allowing it to emerge from the dynamics would require the consideration of a given hillslope process in detail. I hope this becomes clearer with the second numerical example (lines 300– 341).

In some sense "Make it as simple as possible, but not simpler." But what is possible for mountain streams?

I am not sure either, but as far as I know, it is less strict than for AGU journals. To my experience, codes deposited in repositories are not as valuable as it seems unless someone keeps maintaining it contin-In case of OpenLEM I am not uously. even able to provide enough support for a very limited number of users. I prepared a repository and placed it for the moment at http://jura.geologie.unifreiburg.de/esurf-2019-77.zip. Looks as if it takes some time to get a permanent repository at our university this time. And I am also afraid that people will be able to reproduce the results, but not much more.

Best regards, Stefan Hergarten

[revised manuscript text omitted]